# Shifting tasks from pharmacy to non-pharmacy personnel for providing antiretroviral therapy to people living with HIV: a systematic review and meta-analysis

Nyanyiwe Masingi Mbeye,[1,2,3] Olatunji Adetokunboh,[1,2] Eyerusalem Negussie,[4] Tamara Kredo,[1,5] Charles Shey Wiysonge[1,2]

▶ Prepublication history and additional material are available. To view these files please visit the journal online (http://dx.doi.org/10.1136/bmjopen-2017-015072).

For numbered affiliations see end of article.

**Correspondence to**
Dr Nyanyiwe Masingi Mbeye;
nyanyiwembeye@gmail.com

## ABSTRACT

**Objectives** Lay people or non-pharmacy health workers with training could dispense antiretroviral therapy (ART) in resource-constrained countries, freeing up time for pharmacists to focus on more technical tasks. We assessed the effectiveness of such task-shifting in low-income and middle-income countries.

**Method** We conducted comprehensive searches of peer-reviewed and grey literature. Two authors independently screened search outputs, selected controlled trials, extracted data and resolved discrepancies by consensus. We performed random-effects meta-analysis and assessed certainty of evidence using the Grading of Recommendations, Assessment, Development and Evaluation (GRADE) approach.

**Results** Three studies with 1993 participants met the inclusion criteria, including two cluster trials conducted in Kenya and Uganda and an individually randomised trial conducted in Brazil. We found very low certainty evidence regarding mortality due to the low number of events. Therefore, we are uncertain whether there is a true increase in mortality as the effect size suggests, or a reduction in mortality between pharmacy and non-pharmacy models of dispensing ART (risk ratio (RR) 1.86, 95% CI 0.44 to 7.95, n=1993, three trials, very low certainty evidence). There may be no differences between pharmacy and non-pharmacy models of dispensing ART on virological failure (risk ratio (RR) 0.92, 95% CI 0.73 to 1.15, n=1993, three trials, low certainty evidence) and loss to follow-up (RR 1.13, 95% CI 0.68 to 1.91, n=1993. three trials, low certainty evidence). We found some evidence that costs may be reduced for the patient and health system when task-shifting is undertaken.

**Conclusions** The low certainty regarding the evidence implies a high likelihood that further research may find the effects of the intervention to be substantially different from our findings. If resource-constrained countries decide to shift ART dispensing and distribution from pharmacy to non-pharmacy personnel, this should be accompanied by robust monitoring and impact evaluation.

## BACKGROUND
### Description of the condition
By March 2015, 15 million (40.7%) of the estimated 36.9 million people living with

## Strengths and limitations of this study

▶ To our knowledge, this is the first published systematic review reporting the effects of task-shifting from pharmacy to non-pharmacy personnel for dispensing or distributing antiretroviral therapy to patients living with HIV.
▶ The review was written according to the Preferred Reporting Items for Systematic Review and Meta-Analysis (PRISMA) Recommendations for reporting systematic reviews.
▶ The review findings may help to inform antiretroviral therapy guidelines by the WHO.
▶ A limitation of the review is paucity of evidence in this field and use of indirect evidence to inform our results.

HIV (PLHIV) globally were receiving antiretroviral therapy (ART) in low-income and middle-income countries.[1] Combination ART is effective for reducing HIV-related morbidity and mortality as well as preventing HIV transmission.[2] Initiating ART early in the course of HIV infection has been associated with better health outcomes, both at patient and population levels.[3 4] Scale up of ART in low-income and middle-income countries has averted more than 5 million deaths; however, bottlenecks preventing universal access to ART still exist. One challenge is the critical shortage of human resources for health (HRH), including for delivery of essential pharmacy services.

As for pharmacists, although the WHO recommends a minimum of one pharmacist per 2300 population,[5] most countries in low-resource settings such as sub-Saharan Africa do not meet this target. In addition to the absolute shortage, it is likely that there is an uneven distribution of pharmacists in

such settings, as is the case with other specialist healthcare workers who tend to concentrate in urban areas and the private sector, further aggravating the HRH shortage. For instance in South Africa, which is home to the largest number of PLHIV in any country worldwide, in 2010 only 24% of registered pharmacists worked in the public sector where 80% of the population received care.[6]

## Description of the intervention
Studies and programmes report that involvement of pharmacy personnel in HIV care results in improved patient outcomes.[7] For instance, in the USA, the use of a multidisciplinary team approach with pharmacists assuming a central role in the initiation, dispensing and adherence counselling improved treatment outcomes such as viral load, patient retention and medication adherence.[8]

The work of pharmacists includes supply management, dispensing and distributing medications, promoting adherence, identifying and preventing potential medication-related issues (such as over dosage, subtherapeutic dosage, adverse drug reactions, medication errors and untreated indications) and monitoring and reporting adverse drug events.[9 10] In some settings, programmes have implemented alternative models of pharmacy services that shift selected tasks from pharmacy to non-pharmacy personnel. Such alternative models could potentially increase the number of health workers involved in ART distribution, adherence counselling and patient education; free more time for pharmacy personnel; support the integration of ART in primary care settings; minimise the number of facility visits for ART collection; and reduce pharmacy queue waiting times for patients.

The specifics of shifting ART-related tasks from pharmacy personnel have not been addressed in a systematic review. Available systematic reviews on task-shifting focus on clinical services where nurses and non-physician clinicians provide care comparable with physicians.[11] We therefore synthesised the evidence for task-shifting in pharmacy personnel services, where non-pharmacy personnel undertake ART distribution and medication adherence counselling. Table 1 provides definitions for the pharmacy functions distribution and dispensing as they have been used in this review. However, these definitions seem to overlap at facility level.

## How task-shifting using non-pharmacy personnel might work
Within the last decade, several high HIV burden countries adopted task-shifting strategies where nurses and non-physician clinicians initiate and maintain ART. Although this has undeniably expanded access to ART, it is also increasingly essential that long facility waiting times and frequent facility visits to collect ART are addressed to alleviate the burden of care, both for patients and healthcare providers.[9 10]

Recent studies in Uganda, Kenya and Mozambique have shown positive outcomes when non-health professionals (lay people) delivered ART at the community level.[12] In Mozambique, the use of PLHIV for distributing ART, monitoring adherence, reporting outcomes and referring sick patients to health facilities yielded a retention rate of 97.5% among stable patients on ART.[12] In a cluster randomised trial in Uganda, the use of community health workers produced comparable results with the facility-based ART programme in terms of patient retention, viral load suppression and mortality rate.[13] Similar findings were also obtained in Kenya and some other parts of Uganda when lay providers were engaged in ART delivery.[13 14]

Task-shifting has therefore been seen as an achievable solution to the critical human resource shortages affecting scale up of ART.[15] While it is imperative to increase the rate of recruitment and training of health workers as well as improve working conditions to reduce attrition and emigration, the HIV pandemic requires that all possible options are considered to address the critical skills shortage.[16] Such measures may include shifting selected tasks from pharmacy to non-pharmacy personnel, particularly for patients stable on their ART. Task-shifting allows more time for pharmacy personnel to focus on more technical functions such as supply management, pharmacovigilance and patient consultation.

## Why this review is important
Dependence on and shortage of pharmacists are key constraints on ART expansion, but the specifics of task-shifting for ART distribution from pharmacy to non-pharmacy personnel have not been reviewed systematically. Previous systematic reviews of task shifting

| Table 1 | Definitions and descriptions of dispensing and distribution of medications |
|---|---|
| **Term** | **Definition** |
| Dispensing | ► Dispensing is a controlled act that authorises one to select, prepare and provide stock medication that has been prescribed to a patient or client (or his/her representative) for administration at a later time.[29] <br> ► To dispense is to prepare and supply to a patient a course of therapy on the basis of a prescription.[30] <br> ► Dispensing is preparation and distribution of a course of therapy to a patient, with appropriate instructions based on a prescription.[30] |
| Distribution | ► At facility level, drug distribution mainly refers to drugs dispensed by licenced practitioners, such as nurses, doctors, pharmacists or pharmacy assistants and collected by a patient. <br> ► At community level, drug distribution refers to trained lay people collecting prepacked medications from the facility and delivering them to patients with HIV in the community.[29] |

for increasing ART access focused on clinical services where nurses and non-physician clinicians provide care. We systematically reviewed the scientific literature and assessed the efficacy of task-shifting models that use non-pharmacy and pharmacy personnel in distributing ART and assessing adherence to treatment of HIV infection.

### Objective

This review evaluated the efficacy and safety of shifting pharmacy-related tasks, including ART distribution and adherence assessment, from pharmacy to non-pharmacy personnel.

### METHODS

The review was registered in PROSPERO International Prospective Register of Systematic reviews (http://www.crd.york.ac.uk/PROSPERO), registration number CRD42015017034. The protocol as shown in online supplementary file 1 was published in the *BMJ Open*.[17]

### Criteria for considering studies for this review

#### Types of studies

Randomised controlled trials (RCTs) or observational studies with control arms conducted in low-income and middle-income countries.

#### Types of participants

PLHIV receiving ART.

#### Types of interventions

We included studies that evaluated shifting of selected tasks from pharmacy to non-pharmacy personnel. Tasks include distribution, dispensing of ART and adherence to ART as shown in table 1. Pharmacy personnel included pharmacists and pharmacy technicians, while non-pharmacy personnel included patient peer groups, community volunteers, PLHIV, community health committees, nurses, physicians and non-physician clinicians.

#### Types of outcome measures
##### Primary outcome: mortality

Secondary outcomes: virological suppression; number of all-cause sick visits made to the health facility including adverse events; loss to follow-up; adherence to ART (as measured within the study, eg, pill counts, recall methods, digital methods, self report); acceptability to pharmacy personnel, non-pharmacy personnel and patients; and cost and harm including rates of errors.

### Search methods for identification of studies

We performed a comprehensive search of electronic databases and conference proceedings to identify all relevant studies up until 10 May 2016, regardless of language of publication or publication status (published, unpublished, in press and in progress). We searched Cochrane Central Register of Controlled Trials (CENTRAL),

Excerpta Medica Database (Embase), PubMed, ISI Web of Science (Science Citation index) and WHO Global Health Library.

We used appropriate medical subject heading terms, relevant keywords and validated terms for identifying reports of RCTs.[18] Search strategies are reported in table 2. We searched conference abstract archives of the Conference on Retroviruses and Opportunistic Infections, the International AIDS Conference and the International AIDS Society Conference on HIV Pathogenesis, Treatment and Prevention, up to 10 May 2016. We contacted experts and organisations for additional studies.

### Data collection and analysis

Data collection and analysis approach was based on standards from the Cochrane Handbook of Systematic Reviews of Interventions.[18]

### Selection of studies for inclusion

Two review authors (NMM and OA) screened the titles and abstracts of all records to identify potentially eligible reports, and then independently inspected potentially eligible publications for inclusion. Differences were discussed with a third author (CSW or TK) and resolved by discussion.

### Data extraction and management

Two review authors (NMM and OA) independently extracted data onto standardised, prepiloted data extraction forms. The following characteristics were extracted from each included study:

► Study details: citations of publications associated with the study, start and end dates, location, study design characteristics, type of facility involved, investigators, funding sources, recruitment, method of randomisation, sequence generation, method of allocation concealment, blinding of participants and personnel, blinding of outcome assessors, length of follow-up, losses to follow-up, withdrawals or drop-outs and other relevant details.

► Details of the intervention: details of the cadre of health worker, what training or other support or supervision they received and other relevant details.

► Details of participants: trial inclusion and exclusion criteria, numbers of participants entering the trial, sex, clinical staging, CD4 count and other pertinent details.

► Outcome details: definitions of outcomes, details of how outcomes were assessed, numerators and denominators associated with each outcome, completeness of outcome data, effect estimates reported and other relevant outcome information.

### Assessment of risk of bias in included controlled trials

Two review authors (NMM and OA) independently assessed the risk of bias in each study using the Cochrane Risk of Bias Tool, which measures risk of bias in controlled trials across seven domains: sequence generation,

**Table 2** Search strategy for electronic databases

| ID | Search terms |
| --- | --- |
| **PubMed** | |
| #1 | (task*[tiab] OR task-shifting[tiab] OR referr*[tiab] OR referral and consultation[mh] OR role*[tiab]) AND (health personnel[mh] OR doctor[tiab] OR doctors[tiab] OR clinician[tiab] OR clinicians[tiab] OR physician[tiab] OR physicians[tiab] OR 'healthcare provider'[tiab] OR 'healthcare providers'[tiab] OR 'health care provider'[tiab] OR 'health care providers'[tiab] OR pharmac*[tiab] OR apothecar*[tiab] OR chemist*[tiab] OR dispensar*[tiab]) |
| #2 | randomiZed controlled trial[pt] OR controlled clinical trial[pt] OR randomiZed controlled trials[MeSH] OR random allocation[MeSH] OR double-blind method[MeSH] OR single-blind method[MeSH] OR clinical trial[pt] OR clinical trials[MeSH] OR ('clinical trial'[tw]) OR ((singl*[tw] OR doubl*[tw] OR trebl*[tw] OR tripl*[tw]) AND (mask*[tw] OR blind*[tw])) OR random*[tw] OR research design[mh:noexp] OR prospective studies[MeSH] OR control*[tw] OR volunteer*[tw]) OR observational[tw] OR non-random*[tw] OR nonrandom*[tw] OR before after study[tw] OR time series[tw] OR cohort*[tw] OR cross-section*[tw] OR prospective*[tw] OR retrospective*[tw] OR research design[mh:noexp] OR follow-up studies[MeSH] OR longitud*[tw] OR evaluat*[tiab] OR pre-post[tw] OR (pre-test[tw] AND post-test[tw]) NOT (animals[MeSH] NOT human[MeSH]) |
| #3 | (HIV Infections[MeSH] OR HIV[MeSH] OR hiv[tiab] OR hiv-1[tiab] OR hiv-2*[tiab] OR hiv1[tiab] OR hiv2[tiab] OR hiv infect*[tiab] OR HIV[tiab]OR human immune deficiency virus[tiab] OR human immuno-deficiency virus[tiab] OR human immune-deficiency virus[tiab] OR ((human immun*) AND (deficiency virus[tiab])) OR acquired immunodeficiency syndromes[tiab] OR acquired immune deficiency syndrome[tiab] OR acquired immuno-deficiency syndrome[tiab] OR acquired immune-deficiency syndrome[tiab] OR ((acquired immun*) AND (deficiency syndrome[tiab])) or 'sexually transmitted diseases, viral'[mh]) OR HIV[tiab] OR HIV/AIDS[tiab] OR HIV-infected[tiab] OR HIV[title] OR HIV/AIDS[title] OR HIV-infected[title] |
| #4 | (HAART[tiab] OR ART[tiab] OR cART[tiab] OR antiretroviral[tiab] OR anti-retroviral[tiab] OR anti-viral[tiab] OR antiviral[tiab] OR 'Antiretroviral Therapy, Highly Active'[Mesh]) |
| #5 | #1 AND #2 AND #3 AND #4 |
| **Scopus** | |
| | (HIV OR HIV/AIDS OR AIDS OR 'HUMAN IMMUNODEFICIENCY' OR 'ACQUIRED IMMUNODEFICIENCY') AND TITLE-ABS-KEY (TASK-SHIFTING OR TASKSHIFTING OR (TASK* AND SHIFT*) OR TASK* OR (REFERR* AND (NURSE* OR PHARMAC*))) AND TITLE-ABS-KEY (ANTIRETROVIRAL OR ANTI-RETROVIRAL OR ART OR CART OR HAART) AND TITLE-ABS-KEY (RANDOM* OR RANDOMIZED OR RANDOMISED OR TRIAL OR COHORT* OR GROUP* OR COMPAR* OR OBSERVATIONAL OR PROSPECTIVE* OR RETROSPECTIVE* OR 'SYSTEMATIC REVIEW' OR 'META-ANALYSIS') |
| **Web of Science** | |
| | (TS=(HIV OR HIV/AIDS OR AIDS OR 'HUMAN IMMUNODEFICIENCY' OR 'ACQUIRED IMMUNODEFICIENCY') AND TS=(TASK-SHIFTING OR TASKSHIFTING OR (TASK* AND SHIFT*) OR TASK* OR (REFERR* AND (NURSE* OR PHARMAC*))) AND TS=(ANTIRETROVIRAL OR ANTI-RETROVIRAL OR ART OR CART OR HAART) AND TS=(RANDOM* OR RANDOMIZED OR RANDOMISED OR TRIAL OR COHORT* OR GROUP* OR COMPAR* OR OBSERVATIONAL OR PROSPECTIVE* OR RETROSPECTIVE* OR 'SYSTEMATIC REVIEW' OR 'META-ANALYSIS')) OR (TI=(HIV OR HIV/AIDS OR AIDS OR 'HUMAN IMMUNODEFICIENCY' OR 'ACQUIRED IMMUNODEFICIENCY') AND TI=(TASK-SHIFTING OR TASKSHIFTING OR (TASK* AND SHIFT*) OR TASK* OR (REFERR* AND (NURSE* OR PHARMAC*))) AND TI=(ANTIRETROVIRAL OR ANTI-RETROVIRAL OR ART OR cART OR HAART) AND TI=(RANDOM* OR RANDOMIZED OR RANDOMISED OR TRIAL OR COHORT* OR GROUP* OR COMPAR* OR OBSERVATIONAL OR PROSPECTIVE* OR RETROSPECTIVE* OR 'SYSTEMATIC REVIEW' OR 'META-ANALYSIS')) |
| **CENTRAL** | |
| | HIV* OR HIV-1* OR HIV-2* OR HIV1 OR HIV2 OR HIV INFECT* OR HUMAN IMMUNODEFICIENCY VIRUS OR HUMAN IMMUNEDEFICIENCY VIRUS OR HUMAN IMMUNE-DEFICIENCY VIRUS OR HUMAN IMMUNO-DEFICIENCY VIRUS OR HUMAN IMMUN* DEFICIENCY VIRUS OR ACQUIRED IMMUNEDEFICIENCY SYNDROME(AIDS) OR ACQUIRED IMMUNEDEFICIENCY SYNDROME OR ACQUIRED IMMUNO-DEFICIENCY SYNDROME OR ACQUIRED IMMUNE-DEFICIENCY SYNDROME OR ACQUIRED IMMUN* DEFICIENCY SYNDROME in Title, Abstract, Keywords and (TASK-SHIFTING OR TASKSHIFTING OR (TASK* AND SHIFT*) OR TASK* OR (REFERR* AND (NURSE* OR PHARMAC*))) in Title, Abstract, Keywords and ANTIRETROVIRAL OR ANTI-RETROVIRAL OR ART OR cART OR HAART in Title, Abstract, Keywords |
| **WHO Global Health Library** | |
| | (TASK-SHIFTING OR TASKSHIFTING OR (TASK* AND SHIFT*) OR TASK* OR (REFERR* AND (NURSE* OR PHARMAC*)) AND (HIV* OR human immunodeficiency) AND (antiretroviral OR anti-retroviral))) OR (HIV AND task-shifting) OR (HIV* AND task* AND shift*) |

allocation concealment, blinding of participants and personnel, blinding of outcome assessors, completeness of outcome data, selective outcome reporting and other potential biases.[18]

## Measures of effect
We calculated and presented summary statistics for the risk ratio (RR) for dichotomous outcomes with their 95% CIs.

## Unit of analysis issues
The unit of analysis was the individual study participant for all the trials.

## Dealing with missing data
We contacted study authors for one of the included trials to obtain information on the composition of health professionals in the control groups involved in dispensing ART in order to establish whether pharmacy personnel were part of the team. However, the author no longer had those details but assumed that health professionals included pharmacy personnel.

## Assessment of heterogeneity
We used the $I^2$ statistic to measure heterogeneity among the trials. We planned to explore substantial heterogeneity ($I^2$ >50%) by prespecified subgroup analysis. However, there was no evidence of serious heterogeneity.

## Assessment of reporting biases
We minimised the potential for publication bias by using a comprehensive search strategy. As none of the meta-analysis include 10 or more studies, we did not assess publication bias using a funnel plot.[18 19]

## Data synthesis
Meta-analysis was conducted using Review Manager software.[20] We used a random effects model considering the diverse settings of the included studies. Where meta-analysis was not possible, for instance, for adherence and cost, a narrative synthesis of the evidence was carried out. We summarised the certainty of evidence for each outcome using Grading of Recommendations, Assessment, Development and Evaluaton (GRADE).[21 22] The GRADE system defines the certainty of evidence for each outcome as 'the extent of our confidence that the estimates of effect are correct'.[18] The quality rating across studies has four levels: high, moderate, low or very low. Randomised trials are considered to be of high quality but can be downgraded for any of the following five reasons: risk of bias; indirectness of evidence; unexplained heterogeneity or inconsistency of results; imprecision of results; and high probability of publication bias. Similarly, observational studies are considered to be of low quality but can be upgraded for any of these three reasons: large magnitude of effect; all plausible confounding would reduce a demonstrated effect; and the presence of a dose–response gradient. We independently considered the five factors for downgrading the evidence in included trials.

## Subgroup analysis and sensitivity analysis
Sensitivity analyses were not conducted to investigate the effect of excluding studies with high or low risk of bias due to the small number of studies included in the review. No subgroup analyses were indicated by the data.

## RESULTS
A total of 3557 records were identified. Following title and abstract screening, eight studies were identified for full-text screening. Three trials met the inclusion criteria.[14 23 24] The study selection process is shown in figure 1. The characteristics of included and excluded studies are shown in tables 3 and 4, respectively.

Overall, 1993 HIV-infected patients on ART were included. Of these, 1121 were assigned to non-pharmacy personnel model of ART delivery, while 872 were assigned to pharmacy personnel model of ART delivery.

Two studies[14 23] included lay people in the non-pharmacy personnel group, while the third[24] included nurses in the non-pharmacy personnel group as shown in table 3.

## Risk of bias in included studies
All the included trials had low risk of selection bias as a result of adequate randomisation and allocation concealment. Selke et al[14] and Jaffar et al[23] were judged to have low risk of performance bias for blinding because the reported outcomes were objective and unlikely to be influenced by lack of blinding. However, the Selke trial[14] did not give sufficient information to permit judgement of whether detection bias was present. The Silveira trial[24] was judged to have a high risk of performance and detection bias due to lack of blinding for participants, personnel and outcome assessment respectively for all study outcomes. All trials have a low risk of attrition bias as they did not have differential or large numbers of losses to follow-up across the intervention arms. Selective reporting bias was judged to be low for the study by Jaffar et al[23] and unclear for the Selke trial[14] because the trial protocol was not available. The Silveira trial[24] was judged as having unclear risk of reporting bias due to insufficient information and as having a high risk of other biases due to inadequate sample size. Figure 2 provides a graphical summary of the risk of bias assessments.

## Mortality
There were six reported deaths across all three trials. When pooled, we found that there was very low certainty evidence regarding the effect of the task shifting from pharmacy to non-pharmacy personnel on death (RR 1.86; 95% CI 0.44 to 7.95, figure 3), with no significant heterogeneity ($\chi^2$=3.34; df=2; p=0.19; $I^2$=40%). We downgraded the quality of the evidence for indirectness because two of the three trials (Selke[14] and Jaffar[23]) compared complex interventions that included, but were not limited to, pharmacy and non-pharmacy personnel as shown in table 5. There were few deaths, and the effect estimate was

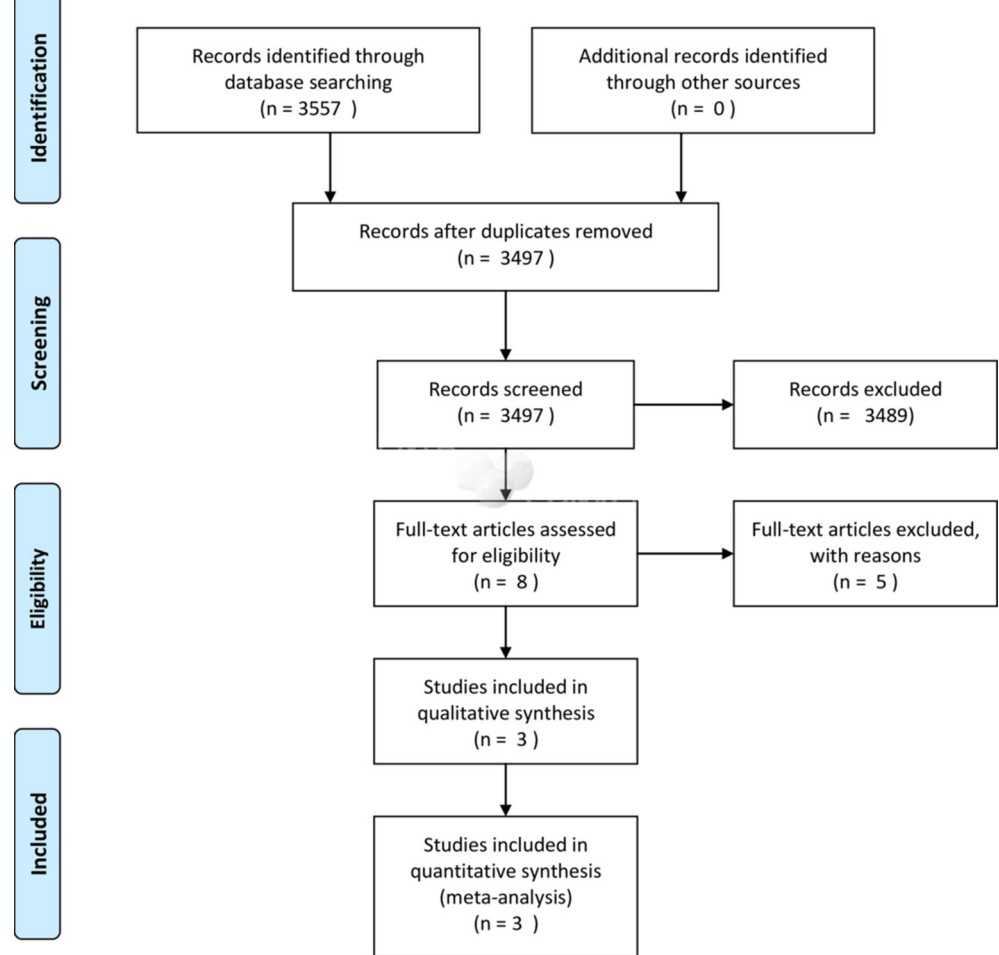

**Figure 1** Flow diagram showing the search and selection of studies.

imprecise, with very wide CIs ranging from appreciable benefit to substantial harm.

### Virological response

We found low certainty evidence that there may be no differences in virological failure between the group cared for by non-pharmacy personnel and that cared for by pharmacy personnel (RR 0.92; 95% CI 0.73 to 1.15, figure 4), with no heterogeneity ($\chi^2$=0.24; df=2; p=0.89; $I^2$=0%). The quality of the evidence for this outcome was low due to serious indirectness and imprecision as shown in table 5.

### Loss to follow-up

We found low certainty evidence that there may be no difference in loss to follow-up between the groups (RR 1.13; 95% CI 0.68 to 1.91), with no heterogeneity ($\chi^2$=0.54; df=2; p=0.76; $I^2$=0%), figure 5). The quality of the evidence was low due to serious indirectness and imprecision as shown in table 5.

### Clinic visits

Selke et al[14] reported significantly more clinic visits in the pharmacy compared with the non-pharmacy personnel group (mean visits 12.6 vs 6.4), p<0.001. Although the non-pharmacy personnel group was found to have fewer

clinic visits, the authors observed that this group attended 64% more clinic visits than originally scheduled. The study by Jaffar et al[23] found a high frequency of outpatient attendance (15 242 visits) in the pharmacy compared with the non-pharmacy personnel group (6691 visits). However, they found similar distribution of new diagnoses between groups where more than 50% were infectious and parasitic infections. The trial by Silveira et al[24] did not report on clinic visits.

### Incidence of opportunistic infections

Selke[14] trial reported an incidence of 13.6 HIV opportunistic infections per 100 person-years in the non-pharmacy personnel group compared with 19.8 HIV opportunistic infections per 100 person-years in the pharmacy personnel group.

### Adherence

All trials reported similar high-level adherence using different measures.

### Cost

In the trial by Jaffar et al,[23] a societal perspective economical analysis showed a higher mean cost per patient per year in the pharmacy personnel group ($838) for health services (staff, transport, drugs, laboratory and clinical services,

**Table 3** Summary of included studies

| Study ID | Setting | Methods | Outcomes |
|---|---|---|---|
| Selke et al[14] | Kenya | **Study design:** cluster randomised trial<br><br>**Description of interventions:**<br>**Non-pharmacy personnel (home-based) group:**<br>Community care counsellors were:<br>▲ Clinically stable patients with self-reported 100% adherence to ART over the 6-month period before recruitment.<br>▲ A patient considered by the clinic staff to be good role model and mentor for other patients.<br>Duties for caregivers during home visits:<br>▲ Obtained and entered data concerning patients' symptoms<br>▲ Vital sign assessment.<br>▲ Assessment of adherence to ART and opportunistic infections prophylaxis.<br>▲ Distributed a 1 month supply of the patients' medications (from a prefilled kit).<br><br>**Pharmacy personnel (facility-based) group:**<br>▲ Nurses/clinical officers or physicians performed the following:<br>– Took an interim medical history.<br>– Addressed any acute concerns.<br>– Reviewed medications.<br>– Prescribed ART and opportunistic infection prophylaxis.<br>▲ Patients collected a 1 month supply of their medication dispensed from the pharmacy. | ▲ Adherence<br>▲ Viral load responses<br>▲ Intercurrent opportunistic infections<br>▲ Hospitalisation<br>▲ Loss to follow-up<br>▲ Change in second-line therapy<br>▲ Mortality |

Continued

**Table 3** Continued

| Study ID | Setting | Methods | Outcomes |
|---|---|---|---|
| Jaffar et al[23] | Uganda | **Study design:** cluster-randomised trial<br><br>**Description of interventions:**<br>**Non-pharmacy personnel (home-based care) group:**<br>▲ Field workers with degree or college diploma who received a week of intensive training at start of study and yearly refresher courses on:<br>– the principles of antiretroviral therapy<br>– adherence support.<br>▲ Field workers visited patients in the homes on motor bikes every month to:<br>– deliver drugs<br>– monitor participants with a checklist of signs and symptoms of drug toxicity and disease progression<br>– provide adherence support<br>– referred patients to physician or counsellor at the clinic when judged necessary.<br>▲ At 2 and 6months after starting therapy, all patients were reviewed by a medical officer at the clinic and then at the 12th month.<br>▲ Drugs were not dispensed during the clinic visits for this group.<br>▲ Patients were visited again when not found at home.<br>▲ Patients asked to come to clinic when unwell.<br><br>**Pharmacy personnel (facility-based) group:**<br>▲ Patients collected drugs every month from the pharmacy.<br>▲ Routine reviews by a medical officer and counsellor at 2 and 3months after start of treatment and every 3months thereafter.<br>▲ Patients assessed by a nurse and referred to a doctor when necessary during clinic visits.<br>▲ Patients followed up at home by field workers when missed an appointment.<br>▲ Patients received vouchers for their households for free voluntary counselling and testing at the clinic.<br>▲ Patients asked to come to clinic when unwell. | ▲ Plasma RNA VL >500copies per millilitre.<br>▲ Either plasma RNA >500copies per millilitre if undetectable at 6months, or an increase of 1000copies between two consecutive tests if RNA detectable at 6months.<br>▲ All-cause mortality.<br>▲ Mortality or plasma RNA VL >500copies per millilitre.<br>▲ Admitted on one or more occasions.<br>▲ All admissions.<br>▲ Death, first admission or change to second-line therapy.<br>▲ Frequency of outpatient attendance.<br>▲ Adherence.<br>▲ Costs of health service delivery and costs incurred by patients to access care. |

**Table 3** Continued

| Study ID | Setting | Methods | Outcomes |
|---|---|---|---|
| Silveira *et al*[24] | Brazil | **Study design:** individually randomised trial.<br><br>**Description of interventions:**<br>**Pharmacy personnel group:**<br>Patients received this intervention onatients received this intervention on a monthly basis through the Da'der method, which consisted a series of scheduled meetings where the pharmacist and patient addressed, reviewed and solved drug-related problems for a 12-month period. Patients received structured counselling from pharmacists on their prescription regimens at the time of their initial drug dispensing and at monthly refill visits.<br>Key elements included:<br>▶ Reviewing the prescription with the patient.<br>▶ Reviewing a card on which medications were colour-coded to facilitate recognition and reduce confusion that might arise from complicated drug names.<br>▶ Reviewing the schedule, length and date of next appointment.<br>▶ Reviewing patient's understanding of the prescription by asking patient to describe it and giving patients verbal information on the expected side effects of their medications and instructing them to seek medical assistance by calling the pharmacist if side effects occurred.<br><br>**Non-pharmacy personnel group:**<br>Patients received usual care (UC) for ART drug distribution. The control group received UC for ART drug delivery. This included the following:<br>At the first appointment, when patients received their medications, a nurse first provided information on:<br>▶ the regimen<br>▶ how and when to use the medications<br>▶ principal side effects<br>▶ importance of adhering to the prescription.<br>Afterwards, patients picked up their medications at a drug-dispensation counter. They had no encounters with a pharmacist. Patients were informed that they could schedule time with a nurse for any questions about their treatment or disease.<br>In both groups, patients were scheduled every 4 months for medical appointments and measurements of viral load and CD4 count. Participants who did not appear for their regularly scheduled appointment were contacted by telephone and asked to return. | ▶ Self-reported adherence calculated by taking the number of tablets that patients reported ingesting and dividing it by the number they should have ingested. Patients were classified as adherent if they reported using 95% or more of the tablets prescribed<br>▶ Depression: this was investigated using a validated Portuguese-language version of the Beck Depression Inventory (BDI) with standard scoring |

ART, antiretroviral therapy; VL, viral load.

**Table 4** Summary of excluded studies

| Study ID | Reasons for exclusion |
| --- | --- |
| March et al[31] | All participants were enrolled in the pharmacist-managed HIV drug optimisation clinic, with no non-pharmacist control group. We excluded this study because we could not make the comparisons due to the absence of a non-pharmacy personnel group. |
| Chang et al[32] | Non-pharmacy personnel did not dispense or distribute ART. |
| Kiweewa et al[33] | Dispensing of ART done in same way for both groups, hence we could not make comparisons. |
| Hansudewechakul et al[34] | Cohort study with comparisons between community and non-community hospitals with both groups having pharmacy personnel. |
| Henderson et al[35] | Observational study with no comparison group. |

ART, antiretroviral therapy.

sensitisation, training and workshops, utilities, supervision and overhead capital) compared with non-pharmacy personnel group ($793). Similarly, total per patient costs to access care per year was higher in the pharmacy personnel group ($54) compared with non-pharmacy personnel group ($18). Costs to access care included transport, lunch, childcare and lost work time. The outcome cost was not prespecified in the protocol; however, this provides additional data that may be relevant for decision makers and was therefore included for consideration.

### Acceptability to pharmacy personnel, non-pharmacy personnel and patients and harm, including error rates.

These outcomes were not reported in the included trials.

### DISCUSSION
### Summary of main results
Two cluster randomised clinical trials and one non-blinded RCT were included in this review of effects of shifting responsibility from pharmacy to non-pharmacy personnel for adherence assessment and dispensing antiretroviral drugs to patients with HIV.

In our meta-analysis, we found very low certainty evidence for the outcome mortality. We are therefore not certain whether the intervention may impact on mortality or not. We found low certainty evidence that there may be no difference in virological response and loss to follow-up between non-pharmacy personnel and pharmacy personnel

| Domain | Trial | | |
| --- | --- | --- | --- |
| | Jaffar 2009 | Selke 2010 | Silveira 2013 |
| Random sequence generation (selection bias) | + | + | + |
| Allocation concealment (selection bias) | + | + | ? |
| Blinding of participants and personnel (performance bias) | + | + | ● |
| Blinding of outcome assessment (detection bias) | + | ? | – |
| Incomplete outcome data (attrition bias) | + | + | + |
| Selective reporting (reporting bias) | + | ? | ? |
| Other bias | + | ? | ● |

● Low risk; ● High risk; ? Unclear risk.

**Figure 2** Summary of risk of bias in included studies.

distributing ART to patients. In addition, we did not find significant differences in adherence to treatment. Selke et al[14] report that the non-pharmacy personnel group had significantly fewer all-cause sick visits to the clinic compared with the pharmacy personnel group. Although the main difference between the Selke[14] and Jaffar[23] trials was the level of education of the non-pharmacy personnel, it did not seem to affect their overall performance.[14 23]

Additionally, the Jaffar trial[23] found the non-pharmacy dispensing strategy to be cost-effective and cheaper to run than the pharmacy dispensing strategy by almost US$45 per patient per annum, and this accounted for only 6% of the total cost of healthcare service expenditure for the intervention group.[23] The patients who were accessing their medications and other healthcare services from pharmacy personnel incurred more costs in terms of transportation, lunch, childcare costs and lost work time. In the first year of the study, the pharmacy personnel group incurred double the cost of healthcare services per patient per annum compared with the non-pharmacy personnel group. This is a challenge in most poor settings in Africa where many cannot afford basic necessities, and this further impedes their access to treatment.[23] Lack of or inadequate financial resources are major factors in late presentation to health facilities, poor access to HIV care and support and low retention in care after initiation of ART.[25 26] Although health services are provided for free in most public institutions, the cost of accessing the services may serve as a hindrance to achieving optimal care and support for HIV-infected patients.[25] The non-pharmacy personnel dispensing ART may reduce the cost of services for both patients and governments with potentially favourable clinical outcomes.

Although issues related to drug–drug interaction, medication errors, prescreening for ART and some special cases of adherence to therapy may be beyond the capacity of non-pharmacy personnel, targeted pharmacy support to non-pharmacy personnel could offer benefits to patients having challenges with their medications especially those who are taking concomitant medications with ART.[27]

This review and meta-analysis has shown that the use of non-pharmacy personnel comprising either lay people or other health professionals such as nurses who are given well-tailored and comprehensive short trainings with

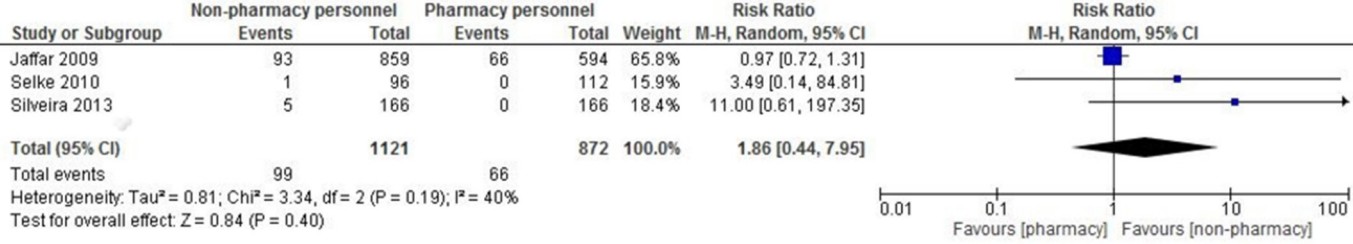

**Figure 3** Effect of shifting dispensing of antiretroviral therapy (ART) from pharmacy to non-pharmacy personnel on mortality.

refreshers may be able to support ART dispensing systems for patients with HIV; however, the low certainty evidence implies that additional trial evidence may change our results. The use of non-pharmacy personnel such as lay people for dispensing ART and monitoring patients' health would enable identification of psychosocial features that might be overlooked by physicians, nurses and pharmacists at the facility. Some of these psychosocial problems such as gender-based violence, food insecurity and alcohol abuse negatively impact on adherence and retention in care.[14] In the case of community-based non-pharmacy personnel, they could also serve as a link between the patients and other healthcare workers and facilitate smooth communication between the patients and the pharmacists on issues such as changes in regimen and dosage of drugs.[14 27]

### Overall completeness and applicability of evidence

Although it may not be possible to rule out publication and language biases in systematic reviews, our electronic search was not restricted to language, setting or publication status.

We identified three studies for inclusion with small sample sizes and event rates. The trials did not specifically pose the question of task-shifting to non-pharmacy personnel and included this approach within complex interventions. As such, this provides indirect evidence where other aspects of the intervention may have resulted in the outcomes reported. The trials did not include some of the outcomes of interest for decision making such as acceptability to participants and feasibility of the intervention. These limitations might have affected our conclusions about the findings from these trials.

### Certainty of evidence

We assessed the certainty of evidence using the GRADE approach.[28] Evidence from this review should be applied with caution considering its very low and low quality. We are therefore not confident enough to state that the

**Table 5** GRADE summary of findings table for pharmacy versus non-pharmacy personnel for dispensing antiretroviral therapy

Population: people living with HIV
Setting: Brazil, Kenya and Uganda (one study per country)
Intervention: non-pharmacy personnel for dispensing antiretroviral therapy
Control: pharmacy personnel for dispensing antiretroviral therapy

| Outcomes (mean follow-up: 12 months) | Pharmacy personnel | Non-pharmacy personnel | Relative effect (95% CI) | No. of Participants (studies) | Quality of evidence (GRADE) |
|---|---|---|---|---|---|
| Mortality | 76 per 1000 | 141 per 1000 (33 to 602) | RR 1.86 (0.44 to 7.95) | 1993 (three studies) | ⊕⊕⊕⊖ very low[1 2] |
| Virological failure | 131 per 1000 | 120 per 1000 (95 to 150) | RR 0.92 (0.73 to 1.15) | 1993 (three studies) | ⊕⊕⊖⊖ low[1 3] |
| Loss to follow-up | 28 per 1000 | 31 per 1000 (19 to 53) | RR 1.13 (0.68 to 1.91) | 1993 (three studies) | ⊕⊕⊖⊖ low[1 3] |

GRADE Working Group grades of evidence.
High certainty: further research is very unlikely to change our confidence in the estimate of effect.
Moderate certainty: further research is likely to have an important impact on our confidence in the estimate of effect and may change the estimate.
Low certainty: further research is very likely to have an important impact on our confidence in the estimate of effect and is likely to change the estimate.
Very low certainty: we are very uncertain about the estimate.
*We downgraded by one for serious indirectness: Two trials compared complex interventions that included, but were not merely limited to, pharmacy and non-pharmacy personnel.
†We downgraded by two for serious imprecision: There are few events and the effect estimates have wide confidence intervals, ranging from appreciable benefit to harm.
‡We downgraded by one for serious imprecision: there are few events, and the effect estimates have wide CIs, ranging from appreciable benefit to harm.
GRADE, Grading of Recommendations, Assessment, Development and Evaluaton; RR, risk ratio.

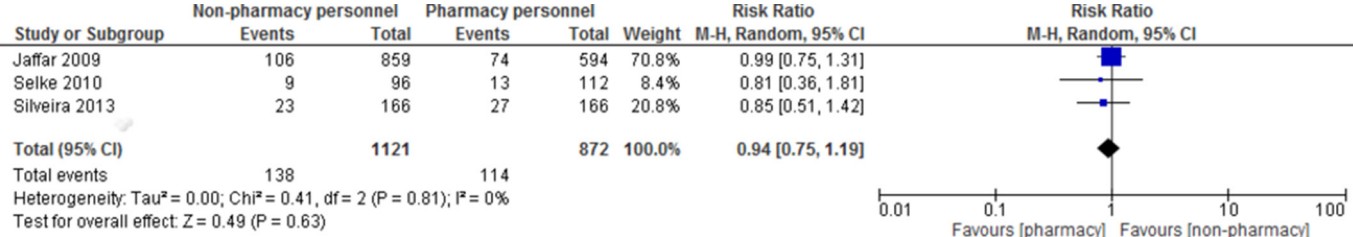

**Figure 4** Effect of shifting dispensing of antiretroviral therapy (ART) from pharmacy to non-pharmacy personnel on virological failure.

**Figure 5** Effect of shifting dispensing of antiretroviral therapy (ART) from pharmacy to non-pharmacy personnel on loss to follow-up.

estimates lie close to the true value, and it is likely that further studies would contribute to the body of knowledge to inform these results.

## CONCLUSIONS

There is low certainty evidence that loss-to-follow up and virological outcomes may be the same whether pharmacy or non-pharmacy personnel provide services to PLHIV on ART in resource-limited settings. This is particularly when accompanied by continued capacity building and supervision by trained pharmacy personnel. However, there is very low certainty evidence regarding the mortality outcome where further evidence will impact the result. The use of non-pharmacy personnel including lay people in dispensing and distributing ART may be cost-effective for the patient and health system. Strong referral systems are crucial in task-shifting of dispensing responsibilities to non-pharmacy personnel to support patients that require advanced medical attention. Due to the critical shortage of human personnel in the health sector, most HIV programmes engaged other cadres of health personnel other than doctors to prescribe and dispense ART at the onset of their programme implementation. This might have contributed to scarcity of studies on the use of pharmacy personnel for dispensing and distributing ART. More trials, with adequate sample sizes to detect clinically relevant health benefits, are therefore needed to investigate task-shifting of dispensing and distributing ART from pharmacy to non-pharmacy personnel. Programmes considering implementation of task-shifting should do so with inclusion of close monitoring and evaluation for health and process outcomes.

**Author affiliations**
[1]Cochrane South Africa, South African Medical Research Council, Cape Town, Western Cape, South Africa
[2]Centre for Evidence-based Health Care, Faculty of Medicine and Health Sciences, Stellenbosch University, Cape Town, South Africa
[3]College of Medicine, University of Malawi, Blantyre, Malawi
[4]Department of HIV/AIDS, World Health Organization, Geneva, Switzerland
[5]Division of Clinical Pharmacology, Faculty of Medicine and Health Sciences, Stellenbosch University, Cape Town, South Africa

**Acknowledgements** The authors would like to thank George Rutherford and Hacsi Horvath (Global Health Sciences, University of California, San Francisco, California, USA) for support and specific assistance in developing the search strategy for electronic databases. Neither the authors' institutions nor the funders played a role in preparing the manuscript, and the views expressed therein are solely those of the authors. We would also like to thank Joy Oliver from Cochrane South Africa, South African Medical Research Council for assistance with the updated search. Both CSW and TK are partly supported by the Effective Health Care Research Consortium. This Consortium is funded by the UK government for the benefit of developing countries (Grant: 5242). The views expressed in this publication do not necessarily reflect UK government policy.

**Contributors** EN, TK and CSW conceived the idea; NMM, TK and CSW drafted the review protocol; NMM, OA, TK and CSW conducted screening, selected the studies, extracted data, performed analyses and drafted the manuscript. All authors reviewed and approved the manuscript.

**Competing interests** None declared.

**Provenance and peer review** Not commissioned; externally peer reviewed.

**Data sharing statement** Additional unpublished data are not available.

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
