## [Reviewer comments · BMJ Open]

ARTICLE DETAILS

TITLE (PROVISIONAL)	Shifting tasks from pharmacy to non-pharmacy personnel for providing antiretroviral therapy to people living with HIV: A systematic review and meta-analysis
AUTHORS	MBEYE, NYANYIWE; Adetokunboh, Olatunji; Negussie, Eyerusalem; Kredo, Tamara; Wiysonge, Charles

VERSION 1 - REVIEW

REVIEWER	Barbara Castelnovo Infectious Diseases Institute Kampala, Uganda
REVIEW RETURNED	18-Nov-2016

GENERAL COMMENTS	I enjoyed reading this manuscript. Task shifting has been a vital component of the scale up of ART in SSA. Few minor comments 1) Table 1 shows the difference between dispensing and distribution. However it is not very clear to me. Distribution at facility levels seems to overlapp with dispensing, at least in the way it is described here 2) The manuscript needs to be revised for typos, there are some words with no space in between. Examples: withbetter, toalleviate, etc Once again this is very important information for resource limited settings, and I recommend publication
--

REVIEWER	Doneal Thomas Envoy Air Inc.
REVIEW RETURNED	25-Nov-2016

GENERAL COMMENTS	A well written and publishable paper that will of interest for researchers in this field. However, figures 3-5 are blurry and some of the texts are difficult to read. I suggest the authors increase the pixels size.
---

REVIEWER	Stephen T. Mennemeyer Ph.D. Professor of Health Economics University of Alabama at Birmingham School of Public Health Department of Health Care Organization & Policy Birmingham Alabama 35294 USA
REVIEW RETURNED	19-Dec-2016

GENERAL COMMENTS

I have reviewed manuscript bmjopen-2016-015072 "Shifting Tasks for Pharmacy to Non-Pharmacy Personnel for Providing Antiretroviral Therapy to People Living with HIV: A Systematic Review and Meta-Analysis" by Mbeye et al.

The paper contains a detail description of the author's methodology for conducting a systematic literature search for scholarly articles on the substitution ("task shifting") of other personnel (e.g. pharmacy technicians, nurses, trained home visitors) for pharmacists in tasks related to dispensing antiretroviral therapy (ART) to persons living with human immunodeficiency virus (HIV). The authors found 3947 articles of which only 3 were of sufficient rigor for meta-analysis. The finding of the meta-analysis is that

"There is low-quality evidence that the use of non-pharmacy personnel including lay people in dispensing and distribution of ART may be as effective as pharmacy personnel performing these functions in resource-limited settings when accompanied by continued capacity building and supervision by trained pharmacy personnel."

The authors are careful and thorough in their methods of review and in their assessment of the strength of the available evidence.

The authors note that costs to access and use care were not in the pre-specified protocol. To their credit, they report on some cost information that they happened to find and they observe (Page 2 lines 40-41) "We found some evidence that costs may be reduced for the patient and health system when task-shifting is undertaken." I suggest that future studies of labor substitution should look for information not only on costs to patients and the health care system but also on comparative wage information such as the wages of the pharmacist and the the substitute personnel.

Specific Remarks

Typo: Table page 22 , line 14 replace "diving" with "dividing"

In Table 4, the article by Hansudewechaku 2012 was excluded because it was a "Cohort study". Why was this particular article excluded when it appears that cohort studies were considered to be worthy of inclusion in the search terms for finding literature as shown in in Table 1? If patients are followed over time and perhaps experience changes in the pharmacy personnel who treat them, couldn't this be either a planned cross over design or an interesting natural experiment? That is, I see no reason a priori to exclude a cohort study. Please explain.

Page 17, lines 344-349 The authors planned to use the I2 statistic "to measure heterogeneity among trials" and to determine when subgroup analyses should be performed. In retrospect, would the authors revise their protocol in light of the recent finding by vonHippel that I2 can be biased in small meta-analyses? This is obviously a rather minor point for the current paper given that only 3 studies suitable for meta-analysis were found but future work should consider this issue.

See: von Hippel, P.T., The heterogeneity statistic I(2) can be biased

	in small meta-analyses. BMC Med. Res. Methodol., 2015. 15: p. 35. With one exception, I have been able to reproduce the calculations of risk ratios and related statistics shown in Figures 3, 4 and 5 using the metan procedure in Stata version 14. The exception is the overall RR for mortality in Figure 3. I do not see how the authors obtain the RR (95% CI) of 1.86 [0.44, 7.95] I find the RR (95% CI) to be 1.01 [0.75, 1.36]. Please verify the reported calculation.
--	---

REVIEWER	Jake Olivier School of Mathematics and Statistics University of New South Wales Sydney, Australia
REVIEW RETURNED	13-Mar-2017

GENERAL COMMENTS	This is a well-done and extremely thorough systematic review and meta-analysis of dispensing ART by pharmacists and non-pharmacists. My primary comment is with regards to mortality. The overall estimate is an 86% increase in the probability of death if ART is dispensed by a non-pharmacist. This includes an 11-fold increase in mortality for one study. I urge the authors to reconsider what they consider to be important, statistical or otherwise. Relying solely on p-values to make such decisions has been shown time-after-time to be faulty logic. The authors can look to the ASAs statement on p-values for some guidance. http://amstat.tandfonline.com/doi/abs/10.1080/00031305.2016.1154108 This guideline is quite relevant to the submission - "By itself, a p-value does not provide a good measure of evidence regarding a model or hypothesis." The authors, at a minimum, should make some judgement regarding the size of the effect. There are articles in literature that discuss what may be considered a large RR. Although I think the size of the RR for mortality needs to be addressed, the authors do an excellent job of discussing the weaknesses/strengths of each study and the overall quality of their estimate. Other issues: The paper, in some ways, is too comprehensive. Was there really a need for a 34 page paper to describe 3 studies? I found some information was repeated like contacting authors. Please define GRADE earlier in the paper. Was MEDLINE used or PubMed? Note that PubMed is a just a means for interfacing with MEDLINE. Do you mean "standardized mean difference" instead of weighted-mean difference? The former conveys more information as the weight is the standard deviation. Could the Hansudewchaku study been included as a sensitivity analysis? Well designed cohort studies can be quite informative,
--

	albeit less than RCTs. . Page 24, line 415. Please clear up citation of "It". Page 28, line 477. P-values are neither 0 or 1. Probably better to write $p > 0.99$ or something similar. How was RR and CIs computed for 0 counts? The usual formulae don't work.
--	---

VERSION 1 – AUTHOR RESPONSE

Reviewer: 1

Reviewer Name: Barbara Castelnovo

Institution and Country: Infectious Diseases Institute, Kampala, Uganda

Please state any competing interests or state 'None declared': None declared

Response: Thank you for this observation. We have included 'None declared' under competing interests on page 34 line 627.

Task shifting has been a vital component of the scale up of ART in SSA.

Few minor comments

Thank you so much for your kind feedback. Below are responses to the comments provided.

1) Table 1 shows the difference between dispensing and distribution. However it is not very clear to me. Distribution at facility levels seems to overlap with dispensing, at least in the way it is described here

Response: We have added a sentence in the paragraph just above table one to clarify. This reads as follows:

“However, these definitions seem to overlap at facility level.”

2) The manuscript needs to be revised for typos, there are some words with no space in between.

Examples: withbetter, toalleviate, etc

Response: The article has been proof read and all typos have now been corrected

Once again this is very important information for resource limited settings, and I recommend publication Response: We highly appreciate this feedback. Thank you so much.

Reviewer: 2

Reviewer Name: Doneal Thomas

Institution and Country: Envoy Air Inc.

Please state any competing interests or state 'None declared': No Competing interest A well written and publishable paper that will of interest for researchers in this field. However, figures 3-5 are blurry and some of the texts are difficult to read. I suggest the authors increase the pixels size.

Response: Thank you for the useful feedback. We have reviewed figures 3-5 and have made the necessary revisions. We have also included 'None declared' under competing interests on page 34 line 627.

Reviewer: 3

Reviewer Name: Stephen T. Mennemeyer Ph.D.

Institution and Country: Professor of Health Economics, University of Alabama at Birmingham, School of Public Health

Department of Health Care Organization & Policy, Birmingham Alabama 35294, USA

Please state any competing interests or state 'None declared': None Declared

I have reviewed manuscript bmjopen-2016-015072 "Shifting Tasks for Pharmacy to Non-Pharmacy Personnel for Providing Antiretroviral Therapy to People Living with HIV: A Systematic Review and Meta-Analysis" by Mbeye et al.

The paper contains a detail description of the author's methodology for conducting a systematic literature search for scholarly articles on the substitution ("task shifting") of other personnel (e.g. pharmacy technicians, nurses, trained home visitors) for pharmacists in tasks related to dispensing antiretroviral therapy (ART) to persons living with human immunodeficiency virus (HIV). The authors found 3947 articles of which only 3 were of sufficient rigor for meta-analysis. The finding of the meta-analysis is that

"There is low-quality evidence that the use of non-pharmacy personnel including lay people in dispensing and distribution of ART may be as effective as pharmacy personnel performing these functions in resource-limited settings when accompanied by continued capacity building and supervision by trained pharmacy personnel."

The authors are careful and thorough in their methods of review and in their assessment of the strength of the available evidence.

The authors note that costs to access and use care were not in the pre-specified protocol. To their credit, they report on some cost information that they happened to find and they observe (Page 2 lines 40-41) "We found some evidence that costs may be reduced for the patient and health system when task-shifting is undertaken." I suggest that future studies of labor substitution should look for information not only on costs to patients and the health care system but also on comparative wage information such as the wages of the pharmacist and the the substitute personnel.

Specific Remarks

Response: Thank you so much for the insightful feedback. Below we provide responses to the comments raised.

Typo: Table page 22 , line 14 replace "diving" with "dividing"

Response: Thank you for this observation. We have made the correction and the sentence now reads: Self-reported adherence calculated by taking the number of tablets that patients reported ingesting and dividing it by the number they should have ingested. Patients were classified as adherent if they reported using 95% or more of the tablets prescribed – Table 3 page 22.

In Table 4, the article by Hansudewchaku 2012 was excluded because it was a "Cohort study". Why was this particular article excluded when it appears that cohort studies were considered to be worthy of inclusion in the search terms for finding literature as shown in in Table 1? If patients are followed over time and perhaps experience changes in the pharmacy personnel who treat them, couldn't this be either a planned cross over design or an interesting natural experiment? That is, I see no reason a priori to exclude a cohort study. Please explain.

Response: Thank you for this comment. Although cohort studies were included in the search strategy, the review complied with the inclusion criteria specified in the protocol. We thought inclusion of cohort studies would be a plus to possibly find RCTs that would probably have been omitted with the RCT search alone.

For the latter point about cross over designs, in the case of having the same group followed over a period of time on the same intervention (pharmacy personnel), we feel it would not meet the definition of cross over design because the patients would have been exposed to the same intervention (pharmacy personnel) throughout and would therefore be difficult to make comparisons on the outcomes.

Page 17, lines 344-349 The authors planned to use the I2 statistic "to measure heterogeneity among trials" and to determine when subgroup analyses should be performed. In retrospect, would the

authors revise their protocol in light of the recent finding by vonHippel that I2 can be biased in small meta-analyses? This is obviously a rather minor point for the current paper given that only 3 studies suitable for meta-analysis were found but future work should consider this issue.

See: von Hippel, P.T., The heterogeneity statistic I(2) can be biased in small meta-analyses. BMC Med. Res. Methodol., 2015. 15: p. 35.

Response: Thank you so much for this comment. We have taken note of the new literature regarding the use of I2 statistic for evaluating heterogeneity. Since the protocol was already published, we will ensure that we take this point into consideration in our future work.

With one exception, I have been able to reproduce the calculations of risk ratios and related statistics shown in Figures 3, 4 and 5 using the metan procedure in Stata version 14. The exception is the overall RR for mortality in Figure 3. I do not see how the authors obtain the RR (95% CI) of 1.86 [0.44, 7.95]

I find the RR (95% CI) to be 1.01 [0.75, 1.36]. Please verify the reported calculation.

Response: Thank you for the comment. We used random effect model and got the result RR (95% CI) of 1.86 [0.44, 7.95]. Your calculation was for fixed effect model which was RR (95% CI) to be 1.05 [0.79, 1.41] using RevMan. We actually stated the right RR.

Reviewer: 4

Reviewer Name: Jake Olivier

Institution and Country: School of Mathematics and Statistics, University of New South Wales, Sydney, Australia

Please state any competing interests or state 'None declared': None declared

Response: Thank you so much for this observation. We have stated 'None declared' under competing interests towards the end of the manuscript on page 34 line 627.

This is a well-done and extremely thorough systematic review and meta-analysis of dispensing ART by pharmacists and non-pharmacists.

Response: Thank you so much for this feedback. We really appreciate. We have provided our responses to the comments raised below.

My primary comment is with regards to mortality. The overall estimate is an 86% increase in the probability of death if ART is dispensed by a non-pharmacist. This includes an 11-fold increase in mortality for one study. I urge the authors to reconsider what they consider to be important, statistical or otherwise. Relying solely on p-values to make such decisions has been shown time-after-time to be faulty logic. The authors can look to the ASAs statement on p-values for some guidance.

<http://amstat.tandfonline.com/doi/abs/10.1080/00031305.2016.1154108>

This guideline is quite relevant to the submission - "By itself, a p-value does not provide a good measure of evidence regarding a model or hypothesis." The authors, at a minimum, should make some judgement regarding the size of the effect. There are articles in literature that discuss what may be considered a large RR.

Although I think the size of the RR for mortality needs to be addressed, the authors do an excellent job of discussing the weaknesses/strengths of each study and the overall quality of their estimate.

Response: Thank you so much for this comment. We indeed found an RR of 1.86 for mortality in our meta-analysis. We concluded that there were no differences between the groups by looking at the 95% Confidence Interval of 0.44 to 7.95 which includes a value of no effect implying that the incidence of mortality is equal between the pharmacy and non-pharmacy personnel group. The p value presented under this analysis is the one for the Chi2 statistic where we are measuring

heterogeneity.

The paper, in some ways, is too comprehensive. Was there really a need for a 34 page paper to describe 3 studies? I found some information was repeated like contacting authors.

Response: Thank you for this comment. We have read the manuscript again. Unfortunately, we have not identified text that could be removed.

Please define GRADE earlier in the paper.

Response: Thank you so much for this comment. We have now included a statement defining the GRADE system which reads as follows on lines 368-370: The GRADE system defines the certainty of evidence for each outcome as “the extent of our confidence that the estimates of effect are correct”

Was MEDLINE used or PubMed? Note that PubMed is a just a means for interfacing with MEDLINE.

Response: Thank you so much for this question. We used PubMed because it is one way to access MEDLINE. We did this because in PubMed, in addition to MEDLINE articles, you will have access to PubMed CENTRAL papers which are full text articles deposited to promote open access and articles that are “in process” that is, prior to being indexed with MeSH terms and articles submitted by publishers “ahead of print”. That is why if you search for the same term in MEDLINE and in PubMed, you may obtain as many as ten thousand more articles in PubMed. More information could be found on this link:

http://www.nlm.nih.gov/pubs/factsheets/dif_med_pub.html

Do you mean "standardized mean difference" instead of weighted-mean difference? The former conveys more information as the weight is the standard deviation.

Response: Thank you for this comment. We decided to use weighted-mean difference on the assumption that all the reported continuous outcomes will be on the same scale.

Page 24, line 415. Please clear up citation of "It".

Response: Thank you for this comment. We have merged the two sentences and now reads as follows: The Silveira trial was also judged as having unclear risk of reporting bias due to insufficient information and judged as having a high risk of other biases due to inadequate sample size. Lines 436-439

Could the Hansudewechaku study been included as a sensitivity analysis? Well designed cohort studies can be quite informative, albeit less than RCTs.

Response: Thank you for this observation. In addition to being a cohort study, the Hansudewechaku trial compared community hospitals to a non-community hospital. In both, there were pharmacy personnel involved in the dispensing of ART hence the study does not provide for our intended purpose of comparing pharmacy and non-pharmacy personnel.

Page 28, line 477. P-values are neither 0 or 1. Probably better to write $p > 0.99$ or something similar.

Response: Thank you for this important comment. However, we have been unable to locate this in the text as specified.

How was RR and CIs computed for 0 counts? The usual formulae don't work.

Response: Thank you for the comments. Inputting the data "0" and others in RevMan produced the results as written in the manuscript. Using Stata metan formulae to analysis the same data also produced the same results.

VERSION 2 – REVIEW

REVIEWER	Stephen T. Mennemeyer Ph.D. Department of Health Care Organization & Policy School of Public Health University of Alabama at Birmingham Birmingham. Alabama 35294 USA
REVIEW RETURNED	24-Apr-2017

GENERAL COMMENTS	The authors have adequately responded to my comments. However, Figure 3 (Effect of dispensing...on mortality) is completely illegible Figure 4 (Effect of shifting dispensing ...l on virological failure) and Figure 5 (Effect of shifting dispensing... on loss to followup) are almost illegible. All three figures need to be uploaded in a legible form.
---

REVIEWER	Jake Olivier School of Mathematics and Statistics University of New South Wales Sydney, Australia
REVIEW RETURNED	13-Apr-2017

GENERAL COMMENTS	The authors have not adequately addressed my concerns. In particular: The authors have apparently used confidence intervals to judge the importance of their relative risk estimates. Making a judgment based on whether or not a null effect is within a confidence interval is mathematically equivalent to basing that decision solely on $p < 0.05$. The ASA's statement "By itself, a p-value does not provide a good measure of evidence regarding a model or hypothesis." is also relevant for interpreting confidence intervals in that manner. The authors have still not considered the size of the effect in their decision making. Personally, if I knew there was an 86% increase in mortality, I would not want drugs dispensed by a non-pharmacy personnel group. I cannot think of any situation where an 86% increase in death would be considered "no effect". It should still be called the "standardized mean difference" because weighted mean difference is unnecessarily vague. What exactly are the weights? The name does not necessarily indicate the weights are the standard deviations. The weights could be constants or some other function of the data. I believe it's a huge problem if the authors cannot indicate what RevMan is doing when there are 0 cell counts. It is not a sufficient response to say "that's what RevMan or Stata gives me, so it must
--

	be correct". It's simply not acceptable to rely on a "black box". The direct quote from page 28, line 477 is: "At study 476 closure, the proportion lost to follow up was similar between the two groups (5.2% for 477 the non-pharmacy personnel group and 4.5% for the pharmacy personnel group), $p = 1.0$." That is the authors have assigned a p-value a number it cannot possibly be. I still think the paper is far too long and I am disappointed the authors could not find any text that could not be removed or at least shortened. Not many people are going to want to read a paper that long describing just 3 studies.
--	---

VERSION 2 – AUTHOR RESPONSE

Reviewers Comments

Reviewer: 4

Reviewer Name: Jake Olivier

Institution and Country: School of Mathematics and Statistics, University of New South Wales, Sydney, Australia

Please state any competing interests or state 'None declared': None declared

Please leave your comments for the authors below

The authors have not adequately addressed my concerns. In particular:

1. The authors have apparently used confidence intervals to judge the importance of their relative risk estimates. Making a judgment based on whether or not a null effect is within a confidence interval is mathematically equivalent to basing that decision solely on $p < 0.05$. The ASA's statement

"By itself, a p-value does not provide a good measure of evidence regarding a model or hypothesis."

is also relevant for interpreting confidence intervals in that manner. The authors have still not considered the size of the effect in their decision making. Personally, if I knew there was an 86% increase in mortality, I would not want drugs dispensed by a non-pharmacy personnel group. I cannot think of any situation where an 86% increase in death would be considered "no effect".

2. It should still be called the "standardized mean difference" because weighted mean difference is unnecessarily vague. What exactly are the weights? The name does not necessarily indicate the weights are the standard deviations. The weights could be constants or some other function of the data.

3. I believe it's a huge problem if the authors cannot indicate what RevMan is doing when there are 0 cell counts. It is not a sufficient response to say "that's what RevMan or Stata gives me, so it must be correct". It's simply not acceptable to rely on a "black box".

4. The direct quote from page 28, line 477 is:

"At study 476 closure, the proportion lost to follow up was similar between the two groups (5.2% for 477 the non-pharmacy personnel group and 4.5% for the pharmacy personnel group), $p = 1.0$."

That is the authors have assigned a p-value a number it cannot possibly be.

5. I still think the paper is far too long and I am disappointed the authors could not find any text that could not be removed or at least shortened. Not many people are going to want to read a paper that long describing just 3 studies.

Responses:

1. The feedback is noted, and we agree that using the p-value or the confidence interval to assign clinical significance is not helpful without consideration of the effect size and other trial related quality indicators. We also note that due to the very low number of events the results of the mortality analysis is very unstable/ fragile.

[ref: Guyatt GH, Oxman AD, Kunz R, Brozek J, Alonso-Coello P, Rind D, et al. GRADE guidelines 6. Rating the quality of evidence—imprecision. *Journal of clinical epidemiology*. 2011;64(12):1283-93.]

We have used the GRADE approach to assess our overall certainty in the evidence. For systematic reviews, quality refers to our confidence in the estimates of effect. On reviewing the mortality results (86% increased mortality ranging from 66% reduction to almost 8 fold increased).

We would like to be more cautious in how we present our results, and suggest we re-GRADE the evidence as very uncertain, based on the high degree of imprecision due to the very low number of deaths in the included trials.

The results (abstract) now state:

We found very low certainty evidence regarding mortality due to the low number of events. Therefore, we are uncertain whether there is a true increase in mortality as the effect size suggests, or a reduction in mortality between pharmacy and non-pharmacy models of dispensing ART (RR 1.86, 95% confidence interval (CI) 0.44 to 7.95, n = 1993, 3 trials, very low certainty evidence).

This has been corrected throughout the results and conclusions of the manuscript.

2. Noted and we agree. There are no continuous outcomes in this review, and we have not needed to make use of mean differences, or standardised mean differences. This has been clarified in the methods and removed.

3. Agreed, the statistical packages are resources to support the analysis. We have used pre-specified criteria stated in the protocol to plan the relevant analyses.

4. Description of the primary studies is reported in the characteristics of included studies table. We have removed the lengthy descriptions of primary studies and therefore there is no reference to the result with the incorrect p-value.

5. We have shortened the review wherever possible, and trust that the more succinct version will be more accessible to readers

Reviewer: 3

Reviewer Name: Stephen T. Mennemeyer Ph.D.

Institution and Country: Department of Health Care Organization & Policy, School of Public Health, University of Alabama at Birmingham, Birmingham. Alabama 35294, USA

Please state any competing interests or state 'None declared': None

Please leave your comments for the authors below

The authors have adequately responded to my comments.

However,

1. Figure 3 (Effect of dispensing...on mortality) is completely illegible

2. Figure 4 (Effect of shifting dispensing ...I on virological failure) and
 3. Figure 5 (Effect of shifting dispensing... on loss to followup) are almost illegible. All three figures need to be uploaded in a legible form.

Response:

We trust the figures are now clear.

VERSION 3 – REVIEW

REVIEWER	Stephen T. Mennemeyer Ph.D. Department of Health Care Organization & Policy School of Public Health University of Alabama at Birmingham Birmingham, Alabama USA
REVIEW RETURNED	04-Jul-2017

GENERAL COMMENTS	The authors have responded adequately to issues that I and others have raised in earlier reviews. In particular, the paper had been edited to improve brevity.
--

REVIEWER	Jake Olivier School of Mathematics and Statistics University of New South Wales Sydney, Australia
REVIEW RETURNED	19-Jun-2017

GENERAL COMMENTS	The authors have adequately addressed my concerns.
--